# OpenReview forum: "CRAKEN: Cybersecurity LLM Agent with Knowledge-Based Execution"
_ICLR.cc/2026/Conference — Submitted to ICLR 2026_

### Official Review · Reviewer_US46 · 2025-10-27

**Soundness:** 2
**Presentation:** 3
**Contribution:** 2
**Rating:** 2
**Confidence:** 5

**Summary:**

This paper presents CRAKEN, a framework for improving LLM performance on cybersecurity tasks, especially CTF challenges. The authors point out two key problems with current LLM agents: they can't access security knowledge beyond their training data, and they struggle to integrate new knowledge into multi-stage planning and execution. The paper tackles this through knowledge-based mechanisms that combine structured and unstructured information.
Main contributions are: a knowledge-enhanced agent architecture with task decomposition, retrieval and injection capabilities; a self-evaluating Self-RAG retrieval process for better output accuracy; Graph-RAG to model entities and relationships using knowledge graphs; a knowledge base with CTF writeups, attack payloads, and code snippets; and empirical evaluation on CTF benchmarks.

**Strengths:**

[+] Comprehensive result analysis including failure type statistics (Figure 5) and retrieval process distribution (Figure 4)

[+] Clear architecture (Figure 1) and module definitions

**Weaknesses:**

[-] Incremental research. Core methods borrowed from existing work (Self-RAG, Graph-RAG)

[-] Limited performance improvements

[-] Inconsistent experimental configurations with insufficient control of variables

[-] Lack of module implementation details

**Questions:**

Q1: Grading module implementation
Section 3 describes three grading components (Relevance Grader, Hallucination Grader, and Solved Grader) in the retrieval pipeline, but I couldn't find implementation details. Are these prompt-based LLM calls with binary classification instructions, fine-tuned classifiers on labeled cybersecurity data, or rule-based heuristics?

Q2: Graph-RAG
How are the queries for Graph-RAG derived? It is not clear how the initial queries are setup, and how do they evolve if retrieval fails. Do they come from the responses from the command executions?

Q3: Hybrid retrieval fusion
Figure 2 shows the hybrid Graph-RAG approach combining structured graph search and vector retrieval, but the fusion mechanism is unclear. Are results from both retrievers simply concatenated or weighted by relevance scores? How do you handle conflicts when graph-based and vector-based results contradict each other?

Q4: Knowledge graph construction
Section 4 mentions the knowledge graph uses semantic triplets like "appeared_in" and "mentions" shown in Figure 2, but critical details are missing. What's the complete relation taxonomy? Is there a formal ontology guiding triplet extraction? How are entities (vulnerabilities, exploits, techniques) identified and canonicalized from unstructured CTF writeups? Is the graph construction fully automated via LLM extraction or does it involve manual curation?

Q5: Inconsistent experimental setup
Table 2 shows several inconsistencies. Why does CRAKEN w/ Graph-RAG only report Claude 3.5 Sonnet results while other configurations test multiple LLMs? This makes it hard to isolate Graph-RAG's contribution from model selection effects. Also, GPT-4.1 performs worse with CRAKEN (11.5%) than D-CIPHER (13.5%), which seems to contradict the framework's premise - what explains this regression? Finally, the paper evaluates DeepSeek V3 but doesn't analyze why this open-source model underperforms so dramatically (2-3% vs 18-22% for commercial models). Do you plan to include more open-source models in future work to verify the system's generalizability?

---

### Official Review · Reviewer_QLh2 · 2025-10-31

**Soundness:** 3
**Presentation:** 3
**Contribution:** 2
**Rating:** 2
**Confidence:** 4

**Summary:**

This paper proposes a knowledge-based LLM agent framework, CRAKEN, which integrates RAG techniques with LLM agents for cybersecurity tasks. Given a Capture-The-Flag (CTF) challenge, the planner-executor based system first decomposes the challenge into multiple sub-tasks and assigns each to an executor. During sub-task execution, each executor is enhanced with knowledge hints retrieved through a Self-RAG framework that employs a hybrid strategy combining traditional vector-similarity search and Graph-RAG methods. Leveraging the retrieved knowledge, the executors solve the challenge. CRAKEN is evaluated on the NYU CTF Bench dataset, which contains 200 CTF challenges from 2017 to 2023, achieving a 3% performance improvement over the state-of-the-art method, D-CIPHER. Additionally, the experiment on different knowledge databases used in the retrieval process demonstrate that step-by-step operational knowledge extracted from CTF write-ups provides superior guidance compared to general or mixed datasets.

**Strengths:**

1. The overall quality of the writing is sufficient.
2. Comprehensive evaluation with different LLMs and knowledge databases. The paper provides a comprehensive evaluation of CRAKEN on four powerful close-source LLMs, including Claude 3.5 Sonnet, Claude 3.7 Sonnet, GPT 4o and GPT 4.1, and a open-source LLM, namely DeepSeek V3. Moreover, the finding derived from the experiments, showing that step-by-step operational CTF write-ups are the most effective knowledge source for the RAG process, is valuable.
3. Clear breakdown of time overhead and cost. In the evaluation result, the paper clearly clarify the average cost per solved CTF and the execution latency of either all attempts and successful cases. The low cost of the framework shows its practicality.

**Weaknesses:**

## The novelty is poor.
The evaluation of integrating RAG with cybersecurity LLM agents on CTF tasks is valuable. However, the work is built upon the existing D-CIPHER framework [1] and directly adopts previously proposed RAG components, namely Self-RAG [2] and Graph-RAG [3]. Although the paper claims to employ an optimized version of Self-RAG, the differences between it and the original implementation described in [2] are not clearly explained. Overall, since the proposed methodology appears to be mainly a combination of existing approaches, the work represents an incremental study with poor novelty.

## Unclear technical details
- The decomposition strategies for task-critical information and the rationale behind their design are not clearly described. Although the contextual decomposition process appears to be a crucial step, its implementation is not thoroughly explained in Section 3 (CRAKEN Architecture). Furthermore, no assessment of the decomposition process’s soundness is provided.
- The criteria for baseline selection are unclear. As shown in Table 1, there are at least seven existing works on LLM agents for cybersecurity. However, the evaluation only considers D-CIPHER and EnIGMA as baselines, without explaining why these two were chosen over the others. This lack of justification raises concerns about potential bias in the evaluation, as the comparison is limited to only two baselines.
- The retry mechanism discussed in Section 5 (RESULTS – Retrieval Process Analysis) is neither detailed in Section 3 (CRAKEN ARCHITECTURE – Retrieval Process) nor illustrated in Algorithm 1. Additionally, as shown in Figure 4, it is unclear why 33.7% of cases in the Retry process reach Success directly, bypassing the Grade Answer step. Further clarification on the design and operation of this retry mechanism is required.

## Insufficient experiments on post-cutoff benchmark
The experiment on the flag-leakage-free cut-off benchmark does not provide strong evidence of CRAKEN's performance persistence on unfamiliar tasks. In the results, there is a notable gap between the 13.64% solve rate on this benchmark and the 21% solve rate on the NYU CTF Bench. While this discrepancy may be partly explained by the difference in dataset sizes (approximately 1:10), the results of D-CIPHER on these 22 post-cutoff cases should also be included. Having this comparison would help demonstrate whether the performance gap between D-CIPHER and CRAKEN remains consistent, thereby supporting the claim that CRAKEN’s superior performance persists even on unfamiliar tasks.

## Difficulty in evaluating the reproducibility
The source code and datasets are not publicly available at the time of review; only the prompts are open-source, which makes it difficult to assess reproducibility.

[1] Meet Udeshi and Minghao Shao and Haoran Xi and Nanda Rani and Kimberly Milner and Venkata Sai Charan Putrevu and Brendan Dolan-Gavitt and Sandeep Kumar Shukla and Prashanth Krishnamurthy and Farshad Khorrami and Ramesh Karri and Muhammad Shafique. D-CIPHER: Dynamic Collaborative Intelligent Multi-Agent System with Planner and Heterogeneous Executors for Offensive Security. https://arxiv.org/abs/2502.10931 \
[2] Akari Asai and Zeqiu Wu and Yizhong Wang and Avirup Sil and Hannaneh Hajishirzi. Self-{RAG}: Learning to Retrieve, Generate, and Critique through Self-Reflection. In International Conference on Learning Representations, 2023. \
[3] Yuntong Hu, Zhihan Lei, Zheng Zhang, Bo Pan, Chen Ling, and Liang Zhao. Grag: Graph retrieval-augmented generation. arXiv preprint arXiv:2405.16506, 2024.

**Questions:**

For detailed major concerns, please see the Weaknesses

1. Please justify the novelty of CRAKEN, and clarify the differences between the optimized Self-RAG used and the original implementation
2. Please explain the design of the contextual decomposition process and include an evaluation on its soundness in the experiment.
3. Please justify the reason of only choosing D-CIPHER and EnIGMA as baselines over other existing works listed in Table 1.
4. Please address the confusion in Figure 4 or clarify the Retry process.
5. Please provide the evaluation result of D-CIPHER on the post-cutoff dataset.

---

### Official Review · Reviewer_49us · 2025-11-01

**Soundness:** 3
**Presentation:** 3
**Contribution:** 1
**Rating:** 2
**Confidence:** 4

**Summary:**

This paper presents CRAKEN, an LLM agent built to complete the NYU CTF Benchmark using retrieval from different knowledge databases. CRAKEN is able to achieve a 3% improvement on NYU CTF Bench.

**Strengths:**

The work does a good job presenting the complex CRAKEN system, and presents a thorough understanding of previous agents built for CTFs and CTF benchmarks themselves. The authors are thorough in their evaluations. They present results for different configurations of CRAKEN, and analyze failure modes and the performance of the graph RAG system.

**Weaknesses:**

Ultimately, this work adds RAG capabilities to a previously presented agentic framework from Xu we al. in order to achieve a minimal increase in performance on the NYU CTF benchmark. The planner-executor based framework is not novel, nor is adding RAG to agents to improve performance. Additionally, despite compiling and presenting a complex system for fetching information relevant to the CTF task at hand, the system only performs 3% more in total on a singular benchmark. This is not nearly a significant enough improvement nor wide enough breadth of evaluations to claim that the agentic framework is state-of-the-art or a worthwhile contribution to ICLR. In fact, under some configurations of the agent performance drops below their baseline comparison of D-CIPHER. This is not to say that negative results should be punished, rather that there is not sufficient breadth of evidence here to show that this system consistently improves performance across the board. Finally, simply evaluating on one CTF benchmark is not enough. CTFs alone are not a reasonable metric for evaluating the cybersecurity capabilities of an agent, and therefore cannot be the sole evaluation metric for a new agent framework. This would be strengthened significantly had the authors presented results for a) other CTF benchmarks, such as Cybench of XBOW's validation set, and b) other, more realistic cybersecurity evaluations, like MHBench, BountyBench, or CVEBench. Currently, the lack of generalizability means that this work is not as strong as it could be.

**Questions:**

1. In table 2, some of the CRAKEN configurations lead to worse performance than the D-CIPHER baseline. Do you have any intuitions as to why an increase in knowledge access results in worse performance?

---

### Official Review · Reviewer_tBDn · 2025-11-03

**Soundness:** 2
**Presentation:** 3
**Contribution:** 2
**Rating:** 4
**Confidence:** 3

**Summary:**

This paper introduces a system for solving cybersecurity Capture-The-Flag (CTFs) that uses language model agents + retrieval of domain knowledge (CTF writeups). On the NYU CTF benchmark, the proposed system improves over the baseline.

**Strengths:**

Cybersecurity LM agents is an exciting area, and using prior domain knowledge is a sensible approach.
The new system (CRAKEN) does improve over prior work.

**Weaknesses:**

The system is rather complex and it is hard to tell which components are most helpful (Table 2 might have athe information but it combines a bunch of variations like model, which is orthogonal to the contributions of the paper). It would be clearer to make clear the three axes of variation: models, scaffolds (RAG or not), and information available to the agent.
Only the NYU CTF dataset is used.  What about Cybench, XBOW, Intercode, CTF-Dojo? The paper would be empirically stronger if it showed the applicability of the method across multiple datasets.
I know this space moves fast, but it would be interesting to see how the latest models (GPT-5, Claude 4.5, etc.) work.

**Questions:**

How do you ensure there is no train-test contamination, especially since you're adding new sources of information (the CTF writeups)?
How were the prompts tuned for each of the different models?

---

### Meta-Review · Area_Chair_RkVk · 2026-01-05

**Summary:**

The paper proposes a cybersecurity LLM agent named CRAKEN. There are concerns regarding novelty, the unclear contributions of each component, the limited evaluation datasets, and the limited improvement demonstrated.

**Reviewer Concerns:**

The authors did not provide a rebuttal. Therefore, these concerns regarding novelty, the unclear contributions of each component, the limited evaluation datasets, and the limited improvement demonstrated remain.

**Reviewer Scores:**

The scores will not be changed, since no rebuttal was submitted.

---

### Decision · Program_Chairs · 2026-01-26

Reject